

# A machine learning-based approach for active monitoring of blades pitch misalignment in wind turbines

Sabrina Milani[1], Jessica Leoni[1], Stefano Cacciola[2], Alessandro Croce[2], and Mara Tanelli[1]

[1]Dipartimento di Elettronica, Informazione e Bioingegneria (DEIB), Politecnico di Milano, Piazza L. Da Vinci 32, Milan, 20133, Italy
[2]Dipartimento di Scienze e Tecnologie Aerospaziali (DAER), Politecnico di Milano, Politecnico di Milano, Via La Masa, 34, Milan 20156, Italy

**Correspondence:** Sabrina Milani (sabrina.milani@polimi.it), Jessica Leoni (jessica.leoni@polimi.it), Stefano Cacciola (stefano.cacciola@polimi.it), Alessandro Croce (alessandro.croce@polimi.it), and Mara Tanelli (mara.tanelli@polimi.it)

**Abstract.** In recent years, timely anomaly detection in wind turbine operations, especially offshore, has become critical. Yet, promptly identifying faults and damages remains a significant challenge, leading to costly maintenance and consuming resources. Rotor blade pitch misalignment is a critical issue, causing downtime and reduced energy production. Traditional inspection methods are resource-intensive, time-consuming, and also struggle to identify the specific misaligned blades. In addition, their accuracy degrades in case of small misalignments and strongly depends on the wind regimes, as they are less reliable in turbulence. The absence of an effective automatic solution persists, requiring costly on-site verification.

To tackle this challenge, this paper introduces a novel machine-learning-based approach for automatic pitch misalignment detection and localization. This approach not only localizes the affected blades but also detects small misalignments as low as 0.1 degrees. The methodology relies on features extracted from a limited set of sensors already integrated into modern wind turbine systems. Specifically, the extracted indicators are designed to effectively integrate frequency and time-domain information on turbine operating conditions, enabling high detection performance even in turbulent wind regimes. The approach is validated across an extended operational envelope using data gathered from a state-of-the-art simulation model commonly used for designing and certifying commercial wind turbine systems.

## 1 Introduction

Persistent vibrations due to misalignment in blade pitches may impact the residual life of a wind turbine leading to system failure through the mechanical degradation of fundamental components such as gearboxes, electronic boards, sensors, motors, and blades (Saathoff et al., 2021). System failure can result in reduced energy production, which is a considerable limitation that requires periodic inspections to monitor the pitch actuator status (Wilkinson et al., 2010). Maintenance interventions on such systems are costly and challenging, especially considering offshore or remotely located machines. Indeed, scheduled maintenance can prove inefficient, resulting in failures between interventions due to sparse intervals or unnecessary interventions from overly short intervals. Consequently, transitioning to condition-based scheduling is pivotal. However, to facilitate this





transition, an automatic diagnostics algorithm is necessary. It would enable continuous health monitoring, promptly detecting early faults and promoting timely interventions to prevent irreversible damages.

Various approaches have been proposed to identify pitch misalignment conditions, typically falling into two categories:
model-based and machine-learning methods. Model-based approaches compare expected and actual turbine behavior, attributing deviations to faults. Machine-learning methods, on the other hand, learn from extensive sets of normal and abnormal operating data to identify general and robust patterns that allow them to distinguish the two conditions. However, regardless of category, these approaches suffer from one or more of these main limitations:

- They require data from sensors that are currently challenging to implement on commercial turbine blades;

- They lose accuracy in turbulent wind conditions or with minor misalignments;

- They only detect overall turbine misalignment without being able to identify the specific misaligned blades;

- They are too complex to allow for an exhaustive interpretation of their decision-making processes, resulting in predictions that are not useful as uncorrelated with the physics of the system.

**Model-based approaches**

In the literature, several works such as Dalsgaard et al. (2009), Tang et al. (2021), Axelsson et al. (2010) and Kusnick et al. (2015) propose model-based techniques to identify and locate pitch and load misalignment anomalies. These techniques, though, tend to be tailored to the specific application as a physical model of the system is required, harming the generality of the method. Furthermore, the proposed methods analyze signals from sensors that could be difficult to install on a real system and are focused more on the analysis of biases and faults of the sensors describing the anomaly of the misalignment of the
blades. In addition, these approaches are presented to be efficiently working in constant wind regime conditions, which are rarely verified in a real scenario. Other model-based techniques have been proposed in Desheng et al. (2021); Cacciola et al. (2016); Cacciola and Riboldi (2017); Cacciola et al. (2018); Bertelè et al. (2018); Bertelè and Bottasso (2022), where the focus is to identify the presence of rotor imbalances and target them through a load compensation technique or via suitable control actions.

These techniques, though, do not propose a method to specifically localize the anomaly or require a modification of the turbine control system, a task practically unfeasible in the case of existing turbines featuring a certified controller.

Conversely, the method presented in this paper is efficient in every wind regime condition, being more general and applicable in a real scenario, where typically turbulent wind conditions are present. In addition, the proposed approach is capable of identifying and locating the presence of the misalignment in a single or multiple blades, without interfering with the nominal
functioning of the wind turbine.





**Machine-learning approaches**

Given the high performance achieved by machine-learning approaches in anomaly detection applications for mechanical systems (Lei et al., 2020), in the last decades, novel methods have been presented in the literature, focused on detecting pitch misalignment (Zhong and Fei, 2022; Kusiak and Verma, 2011). However, such approaches are difficult to implement on a real turbine, as they require several signals rarely available on actual systems. In addition, most approaches, such as the one in Kusiak and Verma (2011), are not capable of estimating the severity of the misalignment nor of locating the blades affected by the anomaly. In our earlier study (Milani et al., 2024), we recognized the potential of machine-learning methods for detecting pitch misalignment. Consequently, we developed a real-time detection method using a machine-learning classifier. Our approach demonstrated strong performance, even in turbulent wind and with minor misalignments. However, while it could classify misalignment levels as small, medium, or high, it lacked localization capability. Additionally, our study only addressed cases where a single blade was misaligned at a time. Conversely, deep learning techniques, such as that discussed in Cacciola et al. (2016), can detect and locate misalignments, even in turbulent conditions, with satisfactory performance. However, their complex structure prevents interpretability, thus hindering their reliability. Indeed, explainability is crucial when dealing with physical systems, as reporting a fault without explaining its physical significance requires operator intervention to understand the situation.

**Novel Contribution**

To overcome state-of-the-art limitations, in this paper we introduce an automatic machine-learning-based diagnostic framework to detect pitch misalignment in wind turbines. Building upon our prior work (Milani et al., 2024), this framework utilizes a minimal set of sensors commonly available in wind turbine systems and employs specialized signal processing techniques to extract relevant features tailored to the physical behavior of the monitored system. Specifically, they have been designed to describe the vibrational behavior and load distribution of the system, as they both are known to be affected in case of misalignment.

Similar to our previous method, real-time feature extraction is performed within a fixed and limited number of rotor revolutions, with subsequent classification using a random forest algorithm to categorize misalignment levels as low, medium, or high.

However, in this work, we extend our dataset to include scenarios with multiple misaligned blades, demonstrating that the approach still achieves high performance. Additionally, we enhance the original framework architecture by designing a hierarchical classification structure. Specifically, after the initial layer, aimed at reporting the presence of a misalignment, our novel methodology integrates an additional layer comprising a random forest classifier to localize the specifically affected blade in case misalignment is detected. Last, to further enhance detection accuracy, we also propose an alternative classification approach for the first layer that leverages linear regression to precisely quantify misalignment severity. Evaluation results validate the effectiveness of our approach, showcasing exceptional accuracy in both misalignment classification, regression, and local-





ization, even under turbulent wind conditions.

To summarize, the significant progress beyond existing solutions in the field are:

85   1. **Precise Detection of Pitch Misalignment**: the presented method is capable of detecting misalignment conditions even in case multiple blades are misaligned, also reporting the exact entity of the misalignment, even when deviations are as minimal as 0.1 degrees.

   2. **Accurate Localization of Affected Blades**: By considering the same minimal set of sensors, the proposed approach is also capable of precisely localizing the misaligned blades. This innovation minimizes the complexity of fault localization, 90   enabling swift and targeted maintenance interventions.

   3. **Interpretability and Robustness**: In our prior research (Milani et al. (2024)), we stressed the need for physics-based features to maintain robustness, even in turbulent wind conditions. This ensures consistent diagnostic accuracy across different operational scenarios, including cases with turbulent winds. Additionally, by combining these features with an interpretable machine-learning technique like random forest and linear regression, we developed a method with an 95   explainable decision-making process. This allows us to rank feature importance, providing insights to domain experts on key indicators influencing the detection process.

The designed approach performance has been validated considering data from an extensive simulation campaign, which encompasses a wide range of different wind conditions and pitch misalignment, affecting both one and multiple blades. To enhance the reliability of the results, we employ a state-of-the-art simulation model commonly used for designing and certifying 100   commercial wind turbine systems.

The article is divided as follows: Section 2 compares the expected physical behavior of a wind turbine affected by pitch misalignment on single or multiple blades to the experimental evidence obtained by analyzing an extensive set of experimental data referred to both nominal and anomalous conditions. Section 3 details the adopted techniques to extract significant and effective features in the time and frequency domain and the designed machine learning methods. Then, the achieved results 105   are presented and discussed in Section 4. At last, in Section 5 the main achievements of the proposed work are underlined and possible future developments are proposed.

## 2   Preliminary Analysis

This Section compares the theoretical and experimental behavior of a wind turbine in healthy and faulty conditions to inspect for informative patterns that can be extracted to quantify the misalignment and locate the affected rotor blades. The available 110   data set is described as well.

### 2.1   Rotor system behaviour

In a balanced rotating system, loads are conveyed to the fixed frame by the rotor only at the harmonics that are multiples of the number $B$ of blades (i.e. for three-bladed rotor only the 3×Rev, 6×Rev, 9×Rev, etc. are transmitted, where "Rev" stands for



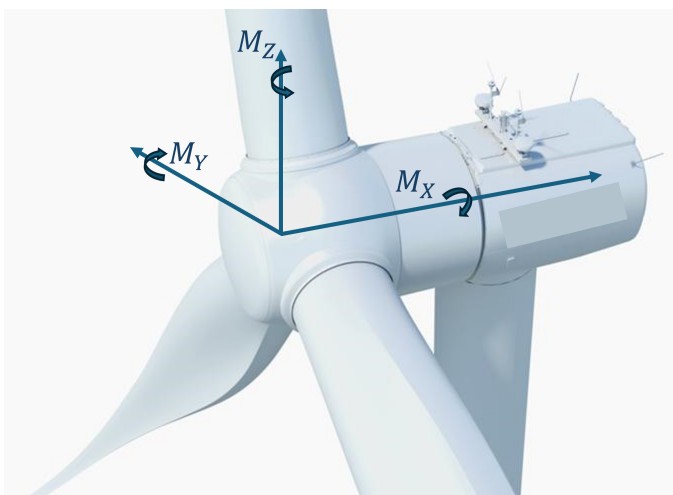

**Figure 1.** Hub coordinate System: $X$ in the direction of the rotor axis, $Z$ upwards perpendicular to $X$, $Y$ horizontally sideways. In the figure the relevant moments $M_y$ (Nodding Moment), $M_z$ (Yawing Moment) are depicted.

the rotor frequency). On the other hand, when the rotor is unbalanced, loads are transmitted at all harmonics, being the $1 \times Rev$
115 the most energetic and deceptive frequency. The most significant signals that reveal the presence of the $1 \times Rev$ harmonic
contribution in a misaligned scenario are the rotor moments in the fixed reference frame along the lateral and vertical axes,
respectively Nodding Moment $M_y$ and Yawing Moment $M_z$, Forces and Accelerations along the same axis. In figure 1 the
mentioned signals of interest in the fixed reference frame are represented.

Given the considered realistic scenario of wind turbines, where typically force sensors are impractical to be installed, the
120 focus in this work is on the blade root bending, nodding, and yawing moment signals, whose sensing is less demanding,
even though the following considerations hold both for accelerations and force signals. As already showed in Cacciola et al.
(2016, 2018); Bertelè et al. (2018) in stationary conditions, the loads exerted on the blades of a balanced three-bladed rotor
are periodic and shifted by 120 degrees, where the period is associated with the rotor speed, $\Omega$. When transmitted to the fixed
frame, blade loads compensate each other such that fixed frame measurements feature only harmonics at frequencies equal
125 to $nB\Omega$, where n is an integer number and $B$ is the number of blades. As soon as a blade shows a pitch misalignment, the
contribution is seen both in the amplitude of the $1 \times Rev$ harmonic conveyed to the fixed reference frame providing insights
into the severity of the misalignment by its amplitude and onto the affected blade by its phase.

When turbulence wind conditions are considered, the blade moments are not exactly periodic. Yet, when a sufficiently long time
window is analyzed, the turbulence effects tend to compensate and balance and only secondary $1 \times Rev$ harmonic contributions
130 can be seen in a healthy balanced rotor.

Therefore, when analyzing the harmonic response of the rotor in the fixed frame, the presence of the $1 \times Rev$ harmonic is a
clear indication of the presence of the anomaly. The amplitude of this contribution is linearly dependent on the severity of the
anomaly, thus the higher the amplitude the more severe the anomaly; and its phase is an indicator of the affected blade.





These considerations hold whenever a fixed wind velocity is considered for different anomaly cases. When considering different
wind velocities, the amplitude of the harmonic response changes, and the higher the velocity the higher the amplitude of the
peak at the $1 \times$ Rev harmonic. In addition, the position of the peak shifts along the spectrum frequencies, according to the
velocities as explained in more detail in the next paragraphs.

## 2.2 Collected Datasets Structure

Before detailing the proposed method, our initial endeavor was to validate the alignment of the turbine's physical characteristics
with their actual operational behavior. To achieve this, we harnessed an extensive dataset gathered from virtual experiments
conducted through a high-fidelity simulation multibody tool (Bottasso and Croce, 2009–2018). Within this simulator, the
flexible elements of the turbine (e.g. tower, blades, shaft) are modeled as kinematically exact beams with predefined $6 \times 6$
sectional stiffness properties (Bauchau, 2011). The classical blade element momentum theory is used for rotor aerodynamics
whereas lifting lines are employed for rendering the aerodynamic forces and moments exerted on blades, tower and nacelle.
The rotor aerodynamic model implemented in `Cp-Lambda` was recently validated against data of an 80-meter-diameter wind
turbine in sheared and yawed inflow conditions (Boorsma et al., 2023).

A multibody model of the NREL 5-MW Baseline was implemented in `Cp-Lambda` and used for all simulations. The turbine
is a realistic three-bladed, variable pitch controlled machine with a diameter of 126 m, that is thoroughly described in Jonkman
et al. (2009).

This simulation environment also possesses the ability to replicate an array of inflow types, including steady, and turbu-
lent wind conditions, and can reliably simulate misalignment scenarios of different severity across one or multiple blades
concurrently.

The analyzed turbulence wind conditions are reproduced according to *Normal Turbulence Model*, NTM, as defined in the
Standards (IEC 61400-1 Ed.3., 2004), with air density $\rho = 1.225 \ [kg/m^3]$. Each simulation set considers 12 conditions at
varying mean wind speeds to cover all operative turbine statuses. In particular, the data set is divided into 6 healthy simulations
and 71 anomaly scenarios. In the anomaly simulations, 7 cases are related to multiple affected blades, and the remaining are
related to single affected blades. The whole dataset is then divided into two main subsets that are used for the classification and
regression analysis. More specifically the regression analysis is performed on data with misalignment ranging from $0.1deg$ up
to $2deg$ and healthy cases as well. More details of the simulations can be found in Tab. 1 and Tab. 2, where the two datasets
exploited for classification and regression are described.

Moreover, in order to gather realistic information, all scenarios, healthy and anomaly, were simulated using turbulent inflows
featuring a different seed.

## 2.3 Spectrum Analysis

The harmonic analysis of the blade moments, when a healthy scenario is considered, exhibits different amplitude peaks ac-
cording to the spectrum harmonics. As stated previously, only multiple of $B$ are conveyed by the rotor on the fixed reference
frame (in the case under study: $B = 3$). Therefore, in the healthy scenario, only harmonics peaks at $3 \times$ Rev, $6 \times$ Rev, $9 \times$ Rev,



| Misaligned Blade(s) | Misalignment angle [deg] | Total number of simulations |
|---|---|---|
| Blade 1 | $\pm(0.1, 0.2, 0.3, 0.4, 0.5, 0.6, 0.7, 0.8, 0.9, 1.0, 1.5, 2)$ | 24 |
| Blade 2 | $\pm(0.1, 0.2, 0.3, 0.4, 0.5, 0.6, 0.7, 0.8, 0.9, 1.0)$ | 20 |
| Blade 3 | $\pm(0.1, 0.2, 0.3, 0.4, 0.5, 0.6, 0.7, 0.8, 0.9, 1.0)$ | 20 |
| Blade 1 and 2 | +0.3, -0.3 | 1 |
| Blade 1 and 3 | +0.6, -0.6 | 1 |
| Blade 2 and 3 | -0.3, +0.3 | 1 |
| Blade 1, 2 and 3 | 0.5, -1, 0.3 | 2 |

**Table 1.** Overall Combinations of Simulations exploited in the "Misalignment Classification"

| Misaligned Blade(s) | Misalignment angle [deg] | Total number of simulations |
|---|---|---|
| Blade 1 | $\pm(0.1, 0.2, 0.3, 0.4, 0.5, 0.6, 0.7, 0.8, 0.9, 1.0)$ | 20 |
| Blade 2 | $\pm(0.1, 0.2, 0.3, 0.4, 0.5, 0.6, 0.7, 0.8, 0.9, 1.0)$ | 20 |
| Blade 3 | $\pm(0.1, 0.2, 0.3, 0.4, 0.5, 0.6, 0.7, 0.8, 0.9, 1.0)$ | 20 |

**Table 2.** Overall combinations of simulations exploited in the "Misalignment Regression"

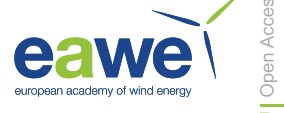 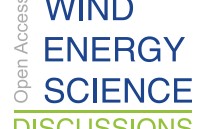

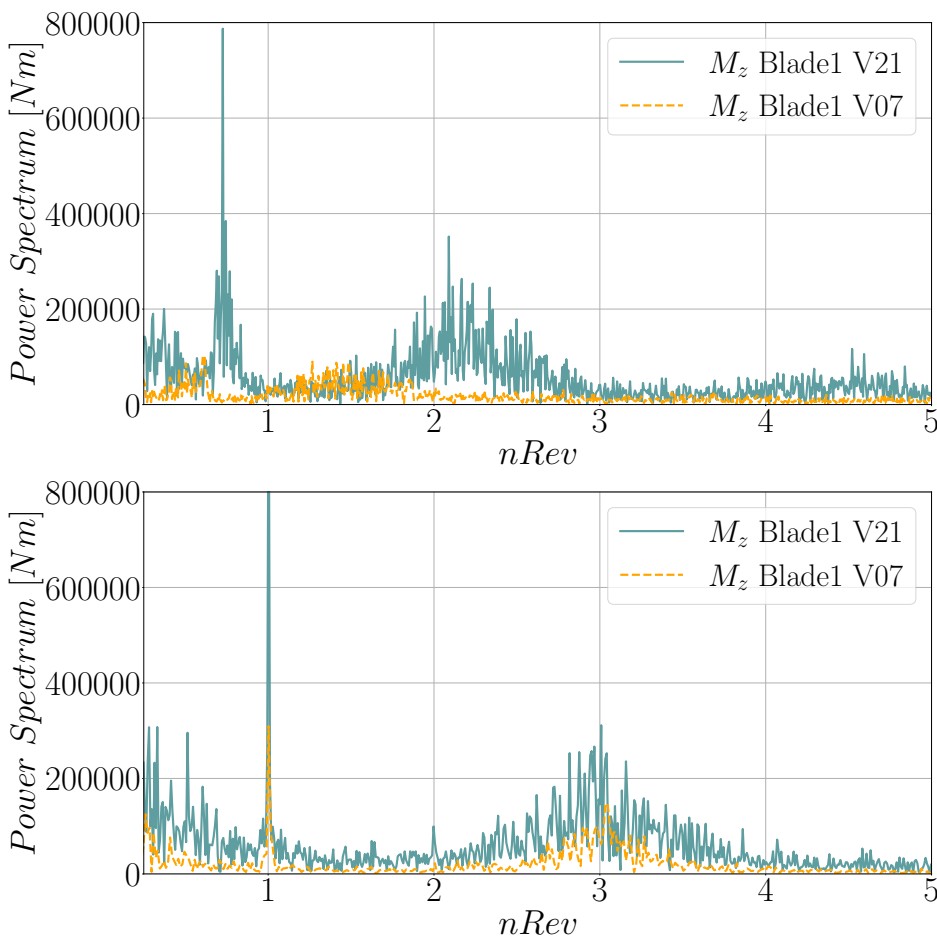

**Figure 2.** In this figure the same signal, namely the Yawing Moment, is analyzed in the frequency domain when $Blade1$ is misaligned. The expected peak at $0.2Hz$ is changing position when considering two different wind speeds in the top graph, while it remains centered around the $1 \times \text{Rev}$ in the bottom figure.

and so on are expected to be seen in the spectrum. As soon as an anomaly arises, the $1 \times \text{Rev}$ harmonics amplitude increases, as shown in the spectrum and the higher the amplitude the more severe the pitch misalignment on the affected blade.

The previous considerations hold in different wind turbulence conditions and at different wind velocities. However, as previ-
ously mentioned, it is necessary to deal with the difference in amplitude and position of the peak in the spectrum due to the rotor velocity. To avoid dependency on the wind velocity, the Fourier Transformation applied to the signals of interest (blades mechanical moments) is performed concerning the Azimuth signal. In such a way, the harmonic response is characterized by a dependency on the rotor frequency revolutions only.

In Figure 2 the same signal, namely the Yawing Moment $M_z$, is analyzed in the frequency domain when considering different
wind speeds ($V = 7m/s$, $V = 21m/s$) and in the case of misalignment on $Blade1$. Indeed, as proved in the figure, when





computing the Power Spectrum using the time-based Fourier Transform, the expected peak at the rotor frequency $f = 0.2Hz$, shifts its position according to the considered wind velocity. Notably, the peak associated with the $1 \times$ Rev frequency at $0.2Hz$ (corresponding to a rotor speed of $6rpm$ at low wind speeds) relocates to higher frequencies at elevated wind speeds. This shift is attributed to the increase in the rotor speed $\Omega$ potentially reaching $0.3Hz$ (namely $9rpm$), under, for instance, nominal

wind conditions. Conversely, when the Power Spectrum is performed with respect to the Azimuth signal, the peak at $0.2Hz$ no longer changes position when considering different wind speeds. Its position in $\times$ Rev remains constant, centered around the $1 \times$ Rev.

In this context, the area subtended below the peak at the $1 \times$ Rev harmonics is a fundamental indicator to distinguish between the healthy and nominal rotor behavior, characterized by a negligible peak amplitude, and the misaligned rotor cases in which

the increment of the peak amplitude leads to higher area values. As reported in Figure 3, the power spectra of the Nodding and Yawing moments were simulated in three different cases: a nominal one and two affected by misalignment, on a single and three blades at a time. Each case is simulated for the same wind velocity (a medium wind speed $v = 15m/s$). The healthy case is characterized by no peak at the $1 \times$ Rev, while the other two cases show a peak whose amplitude has a dependency on the anomaly magnitude. Indeed, signals reported in the figure refer to the following cases:

– case where only $Blade$ 2 is affected by a misalignment of $0.5deg$ in light blue

   – case where all blades displaced by $1.0deg$, $-0.5deg$ and $0.3deg$ respectively, reported in darker blue and dashed line.

Indeed, for this latter case, the peak has a greater amplitude than the former one, as proved in the figure.

Moreover, as stated before, the phase of this harmonic is an equivalently important indicator for the location of the pitch misalignment, hence the single or the multiple blades affected by the anomaly. The phase has been computed by employing a

one-revolution moving window, to demodulate sine and cosine components at the $1 \times$ Rev component of the signals. Subsequently, a 10-minute averaging process has been applied to both components and then the phase was determined by calculating the *arctangent* function on the averaged sine and cosine amplitudes.

In Figure 4, the $1 \times$ Rev phase harmonics are displayed for both Nodding and Yawing Moments in each case. As reported in the figure, there are 10 clusters that can easily grouped. Phases of the cases where $Blade$ 1 is affected by a negative

misalignment are separated into two different groups on the graph. However, they can be considered as a single one as they differ in location on the graph according to the sign of $M_y$ signal phase. There are two clusters associated with each single misaligned blade (both for positive and negative pitch misalignment). Additionally, one cluster is dedicated to cases in which two blades are misaligned simultaneously and lastly, one final group encompasses cases in which all blades are misaligned. Notably, when two blades are simultaneously misaligned, their phase clusters are positioned between the phase clusters related

to the individual misalignment of blade cases. In other words, the phases of the cases in which $Blade$ 1 and $Blade$ 2 are misaligned, fall between the phases of cases in which only $Blade$ 1 and only $Blade$ 2 are misaligned according to the sign of the misalignment. Hence, the area subtended the peak at the $1 \times$ Rev harmonics for both Nodding and Yawing moments are paramount indicators to be computed as well as the phase of these harmonics.

As a further contribution to the isolation of the affected blade, an additional analysis is presented. Considering the blade

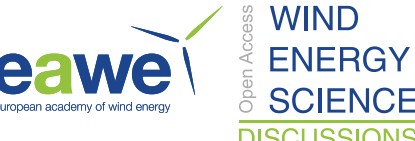

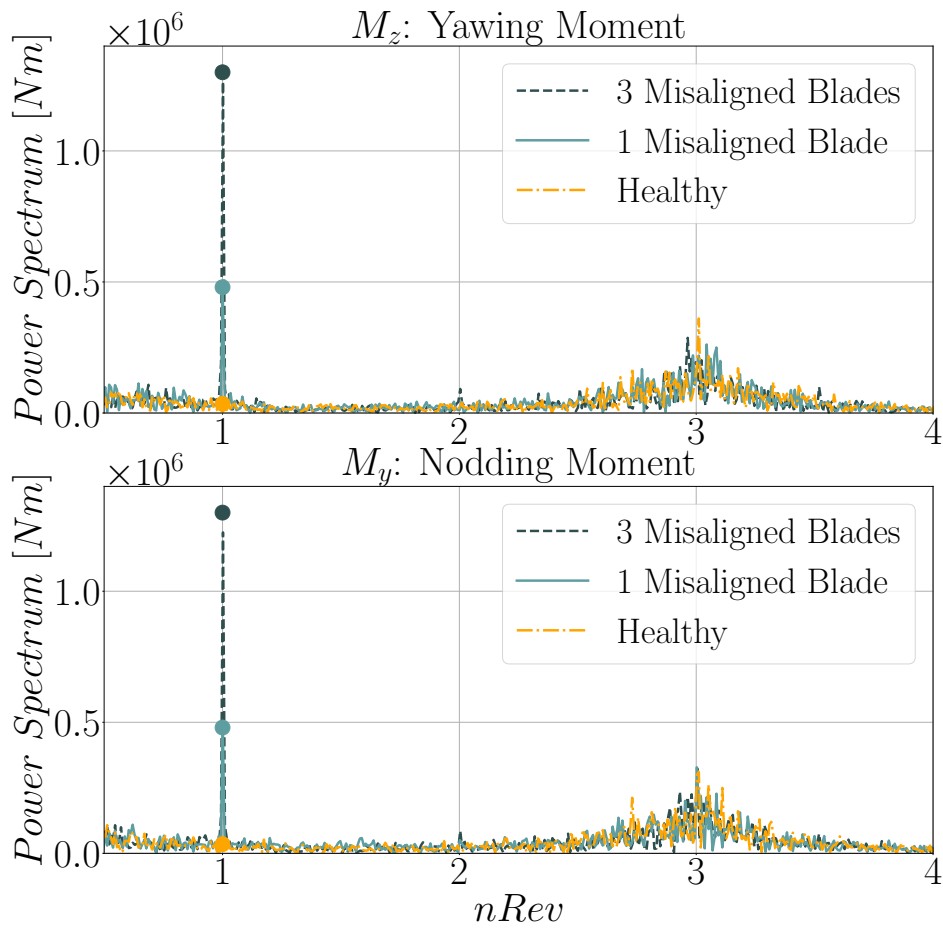

**Figure 3.** In this figure the mechanical moments signals spectrum are analyzed in the frequency domain in the case of a Healthy and two anomaly cases. The peak at the $1 \times$ Rev is present only in the anomalous cases.

moments signals on the rotating reference frame when a balanced rotor scenario is considered, the moments are periodic and overall exhibit similar behavior, oscillating around the same average value. As soon as a single blade is affected by the misalignment, the anomalous blade moment outdistances the other signals. The higher the misalignment, the more significant the distance with the other signals. In addition, the direction towards which the anomalous signal shifts, is an indicator for the misalignment sign (*i.e.* positive or negative misalignment). The same considerations hold for the cases when two or more

blades are affected by the misalignment, as the overall distribution of the signals in the time domain changes concerning the expected behavior.

Therefore, the average values of the blade moments in the time domain are important indicators as well to identify the affected blades. Furthermore, in case more than one blade is affected, the distance between average values is needed as well.



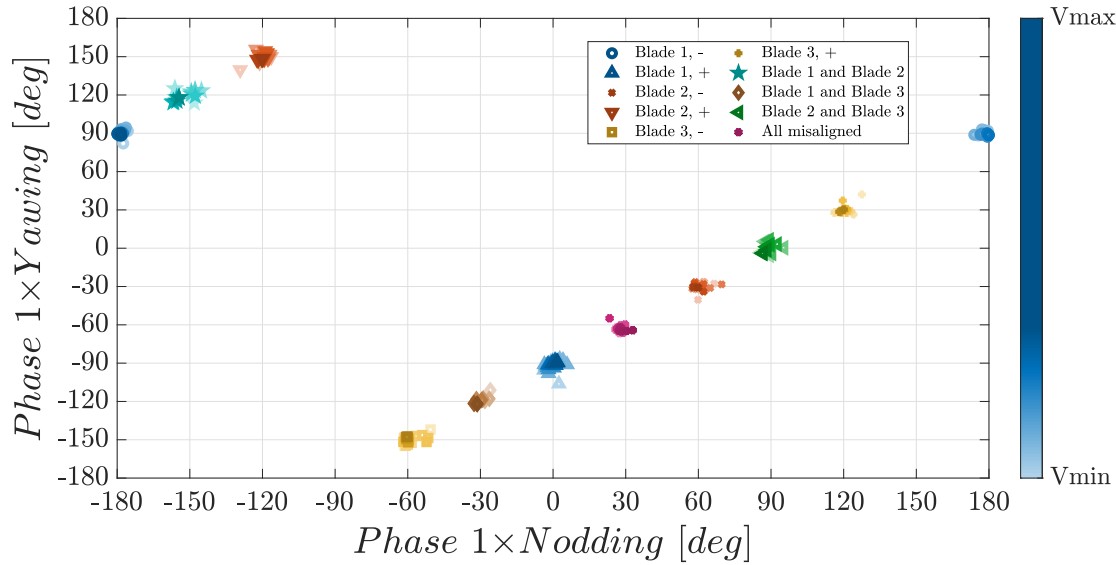

**Figure 4.** In this figure the mechanical moments signals $1 \times$ Rev harmonics phases are reported. Cases in which the same blade is affected by the anomaly are represented with the same color, while a different marker for the same color represents a different misalignment sign for the same blade. Different instances of the same symbol with the same color shading from a lighter to a darker tone represent different phases for the same case but considering different wind velocities. Specifically, the wind velocity increases from the minimum to the maximum considered speed $V_{min} = 3 m/s$ - $V_{max} = 25 m/s$ as the bar on the right reports. For the sake of conciseness, the bar is only reported for the case in which blade 1 is misaligned, the same meaning holds for the other cases and colors.

### 2.4 Features Extraction

Given the insight derived from the comparative analysis of the physical system behavior in healthy and misaligned conditions, we define a set of physically informative features to extract from each available simulation. Specifically, we considered:

1. Area subtended below the peak at the $1 \times$ Rev, both for Nodding and Yawing Moments: $A(M_y)$, $A(M_z)$;

2. Phases of these harmonics: $\phi(M_y)$, $\phi(M_z)$;

3. Average values of the blade moments in the rotating reference frame and in the time domain: $M(M_y)$, $M(M_z)$;

4. the absolute difference of the average values $\Delta M_{i,j} = M_i - M_j$, where $M$ stands for the mean values of the considered signal and i, j are the considered blade (e.g. i=1,2,3 and j=1,2,3).

As discussed in Section 1, we choose to extract the features using a moving window approach to enable real-time classification. To determine the optimal window size, we conducted a fine-tuning procedure, selecting a fixed number of rotor revolutions $N$ instead of a time duration. This novel approach aligns with our decision to perform the Power Spectrum analysis based





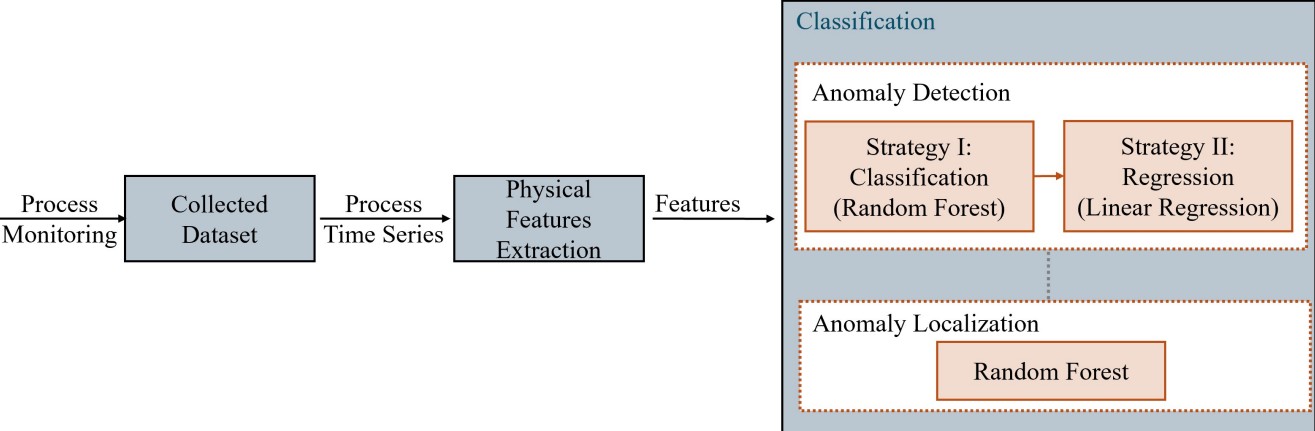

**Figure 5.** Hierarchical Detection and Localization Architecture

on the Azimuth signal. Proper sizing of the window is crucial to prevent extracting irrelevant features. Thus, we perform fine-tuning taking into account both the simulation lengths (600 seconds each, according to industrial standards (IEC 61400-1 Ed.3. (2004))) and wind velocity ranges, resulting in an optimal window size of 40 Azimuth revolutions. This size ensures robust Fourier transform performance across varying wind speeds. Along the size of the chosen window, a sliding factor of 1 Azimuth revolution has been selected. In this approach, the sliding factor determines the frequency of shifting the analysis window along the dataset. A smaller sliding factor, such as one revolution, facilitates a more detailed examination of the data, capturing

fine-scale variations, while ensuring a continuous and comprehensive analysis.

## 3  Hierarchical Misalignment Detection and Localization Classification Architecture

This Section outlines the hierarchical classification architecture we designed for detecting and isolating pitch misalignment. As described in Section 1, our classification pipeline features two layers: the first identifies misalignment, categorizing its severity

as low, medium, or high using the classification model, or precisely predicting the amount using the regression one, based on user preferences. The second layer, given that a misalignment is detected, localizes the affected blades. Both layers rely on random forest models for classification, and linear regression for the regression. These models have been chosen for their robustness, accuracy, and interoperability. Specifically, this algorithm combines the output of multiple shallow binary tree clas-sifiers, which splits data according to specific criteria, minimizing the differences in the child nodes at each iteration. Accuracy,

precision and Gini index (Ceriani and Verme (2012)) are the most common criteria taken into account when performing the splitting procedure. This structure makes random forest models extremely robust and effective, representing the state-of-the-art considering bagged-trees-based classification techniques. In Figure 5 a scheme of the employed hierarchical architecture is proposed.





### 3.1 Pitch Misalignment Detection

As previously discussed, the first classification layer aims to identify misalignment conditions using the features detailed in Section 2. Specifically, in this layer are considered as features the area subtended below the peak in the spectrum of the previously described signals $(A(M_y), A(M_z))$

#### 3.1.1 Misalignment Classification

In our prior study (Milani et al. (2024)), we train a random forest classifier to learn patterns within the given features to assign
each instance, consisting of a set of features corresponding to 40 turbine revolutions, to one of four misalignment conditions:

- *0 deg misalignment*: Healthy cases

- *0.1 deg ≤ p < 0.6 deg misalignment*: Anomaly Case with low misalignment p

- *0.6 deg ≤ p < 1.5 deg misalignment*: Anomaly Case with medium misalignment p

- *p ≥ 1.5 misalignment*: Anomaly Case with high misalignment p

where p is the actual value of detected misalignment. Specifically, this model was composed of 4 trees, each with depth 10.

#### 3.1.2 Misalignment Detection

In this study, we expand upon previous research by introducing and evaluating the performance of a linear regression model. This model, trained using the same set of features employed in the misalignment classification, offers precise quantification of misalignment severity considering default hyper-parameters.
Similar to the random forest classifier, linear regression provides an interpretable decision-making process, enabling users to understand how the model determines the extent of misalignment. However, choosing this regression model instead of the classifier offers users an additional tool for accurately assessing misalignment severity, particularly when precise knowledge of misalignment extent is required. However, it is important to notice that classification may be enough unless continuous monitoring of progressive misalignment evolution is required.

### 270 3.2 Misaligned Blades Localization

If a misalignment condition is detected, the instance is passed to the second layer, where another random forest classifier is employed to localize the affected blades. This classifier considers additional features specific to that window frame, specifically the average values of the mechanical moments and their distances ($\Delta M_{i,j}$ and M($M_y$), M($M_z$)). These indicators have been identified as crucial based on domain expertise and analysis discussed in Section 2.
In this case, the possible output classes are:

- *Blade 1* misaligned





- *Blade 2* misaligned

- *Blade 3* misaligned

- 2 Blades simultaneously misaligned

– All blades simultaneously misaligned

In this second instance of Random Forest, a different selection for the hyper parameters has been set, in particular the number of trees has been set to 10 with max depth 7. Given the higher complexity of this second task, a larger number of trees has been set with respect to the previous instance of Random Forest Classifier (the higher the number of threes the better in term of accuracy, yet higher computational time is expected, increasing with the depth of the trees). Therefore, 10 trees with depth 285   7 has been chosen as a trade off between accuracy and computational time.

## 4    Results Evaluation and Discussion

In this Section, the experimental results obtained applying the hierarchical classification architecture described in Section 3 to the dataset illustrated in Section 2 are discussed. To ensure robustness and consistency of the reported results, we adopt an hold-out-validation approach, using 70% of the dataset as training set and the remaining 30% for the validation. The dataset 290   has been randomly divided. In evaluating our approach, we rely on commonly employed metrics in the machine-learning field. Specifically, for classification, we consider precision, recall, and F1-score. Precision measures the accuracy of predictions within a specific class, while recall (or sensitivity) assesses the proportion of correctly predicted instances within that class out of all instances belonging to the class. Last, the F1-score combines recall and precision, providing a balanced measure that is often regarded as the harmonic mean of the two metrics. Additionally, we report the support, representing the number of 295   instances used to compute the metrics. On the other hand, for evaluating the performance of the regression model, we employ the root mean square error (RMSE), which quantifies the deviation between predicted and actual misalignment for continuous output.

### 4.0.1    Misalignment Detection Results

Initially, we extract the features described in Section 2 from the dataset and we provide the resulting instances into the first 300   classification layer. The outcomes provided by the classification model are illustrated in Fig. 6, depicting the normalized area under the peak in the spectra for the test instances and comparing their true and predicted class of misalignment. The normalization scales the data based on the minimum and maximum values within each est of wind velocities from 0 to 1. The presented results indicate that the majority of instances have been correctly classified. These outcomes are further described in Table 3, which highlights the high performance of the first layer based on key metrics in the field, including precision, recall, 305   and F1-score. Specifically, the approach achieves an average F1-Score of 93.0%.

Concerning the linear regression model, Figure 7 compares the true and predicted misalignment values. The results demonstrate the accuracy and reliability of the regression model, suggesting it is a viable alternative to the classification model when

**Figure 6.** Fault and quantification output, in the top left figure the healthy cases in the data set and the predicted ones are reported, while in the other figures low, medium and high misaligned cases are reported, considering both the misaligned cases in the data set and the predicted ones.

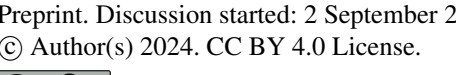



| Classification Report | | | | |
|---|---|---|---|---|
| **Label** | **Precision** | **Recall** | **F1- score** | **Support** |
| Healthy | 0.95 | 0.94 | 0.95 | 119 |
| Low | 0.92 | 0.91 | 0.93 | 555 |
| Medium | 0.93 | 0.95 | 0.94 | 539 |
| High | 0.93 | 0.87 | 0.90 | 57 |

**Table 3.** Metrics computed from the fault and quantification output, precision focuses on the accuracy of positive predictions, recall emphasizes the ability to capture all positive instances, the F1-score balances precision and recall and support provides context by indicating the frequency of each class in the dataset.

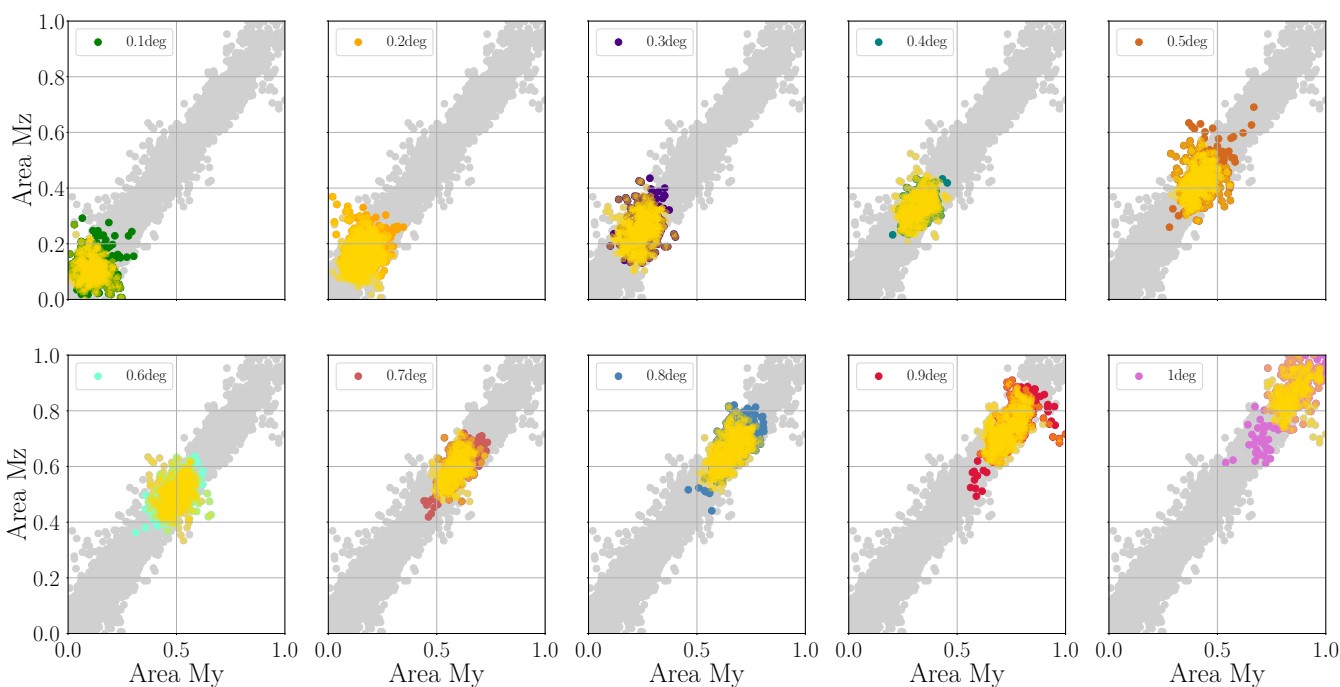

**Figure 7.** Fault and quantification output, linear regression: in each plot a different misalignment instance is considered.

a more precise diagnosis of misalignment severity is needed. Additionally, Figure 8 displays the distribution of predicted misalignment values for each degree of true misalignment. Notably, the precision of the regression is remarkable. The average

values of the boxes closely align with the diagonal of the graph, indicating that predicted misalignments closely match actual values. Furthermore, the separation of values at the $25^{th}$ and $75^{th}$ percentiles in each boxplot demonstrates the model's ability to differentiate between various misalignment levels.




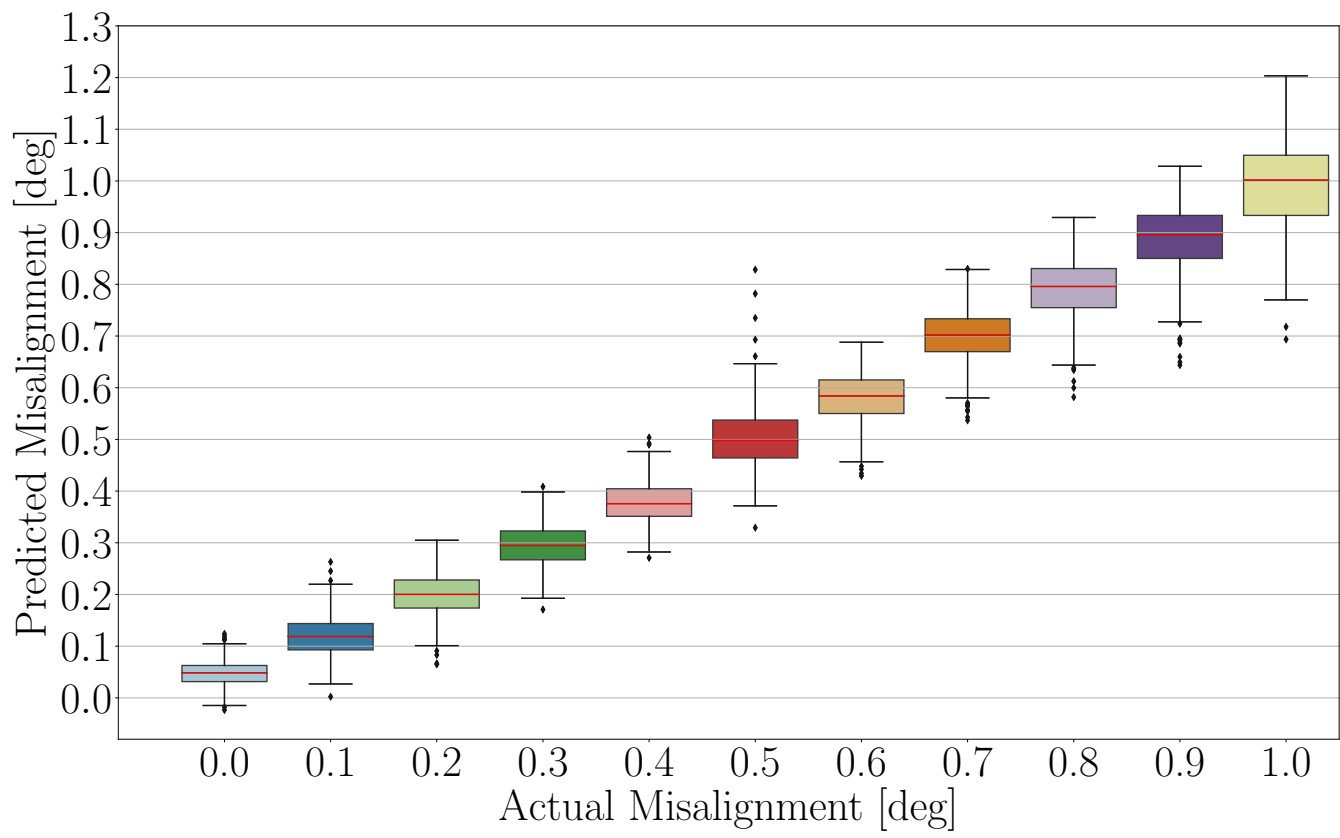

**Figure 8.** Fault and quantification output, linear regression: boxplot distribution for nominal and anomaly cases.

### 4.0.2 Misaligned Blades Results

Once misalignment detection is completed, the next step is to localize the misaligned blades. Figure 9 illustrates a comparison
between the true condition and predictions made by the second layer random forest classifier. Instances are reported in the
three-dimensional feature space, with each point representing a specific instance based on the distribution along the Nodding
Moment average values distances ($\Delta M_{12}$, $\Delta M_{13}$, $\Delta M_{23}$). These results prove the effectiveness of our approach in accurately
localizing the misaligned blades. Additionally, Table 4 provides a quantitative overview of these results, presenting metrics
computed for the second classification phase. Specifically, the average F1-score assessed by the model is 97.0%.

To provide further insights concerning the predictive process underlying the isolation phase, Figure 9 reports different
perspectives of the features space in the considered cases, along with their actual class of belonging. Notice that, the cases
where two blades are simultaneously affected by misalignment are placed in the plot in between the related single misaligned
blades distribution. In other words, as reported in the Figure the points referred to the case in which Blade 1 and Blade 3
are simultaneously misaligned is in between the cases of single misaligned Blade 1 and single misaligned Blade 3. Notice
that, this behavior is consistent with the considerations previously reported on the phase analysis illustrated in Figure 4, where

| Classification Report | | | | |
|---|---|---|---|---|
| **Label** | **Precision** | **Recall** | **F1-score** | **Support** |
| Blade 1 | 0.99 | 0.99 | 1 | 188 |
| Blade 2 | 1 | 0.99 | 1 | 77 |
| Blade 3 | 0.99 | 1 | 0.98 | 94 |
| Two Blades | 0.91 | 0.91 | 0.94 | 20 |
| Three Blades | 0.94 | 0.96 | 0.97 | 90 |

**Table 4.** Metrics computed from the fault isolation output.

phase distributions of cases in which two blades are misaligned are placed in between the related phases groups of the single misaligned blades phases. This study significantly advances the current understanding of pitch misalignment in wind turbines by addressing a critical aspect that has been conspicuously absent in the existing literature. Notably, the issue of misalignment in multiple blades has been historically overlooked in prior research. Our findings bring forth a pioneering perspective, shedding
light on the complexities associated with this previously unexplored dimension. By introducing a comprehensive analysis of misalignment in systems with multiple blades, this research fills a notable gap in the current body of knowledge and provides valuable insights that are crucial for industrial applications as well.

## 5    Final Considerations and Future Work

In this work, we propose an automatic machine-learning-based framework for detecting pitch misalignment in wind turbines.
Using a minimal set of signals and processing techniques, our framework extracts features tailored to the system's behavior, ensuring robustness across various turbulence conditions and promoting the interpretability of the decision-making process. In more detail, the designed hierarchical classification architecture employs, in the first layer, either a random forest classifier or a linear regression model to detect misalignment. Upon detection, in a second layer, a random forest classifier identifies the misaligned blades.
Validation using data simulated from a state-of-the-art simulator confirms the approach's ability to accurately detect and localize misalignment, even with multiple misaligned blades and in different turbulence conditions, achieving an F1-score exceeding 93%. Regression outcomes analysis also proves the framework's capability to detect misalignments as low as 0.1 degrees with a root mean square error of 5.48%.

While we recognize the reliability of the simulator used for simulating the data, we acknowledge that a limitation of our
work lies in the lack of validation on real data. Therefore, future work will concentrate on assessing the system performance using data gathered from operational turbines, despite the associated challenges, such as the lack of misalignment occurrences and the difficulty in accessing turbine data due to their remote locations. Indeed, we believe that this validation is crucial for assessing the framework's robustness to real-world conditions. At the same time, we are expanding the simulated data

**Figure 9.** Fault Isolation output, different perspectives. When two blades are simultaneously misaligned (light blue points in the plot), they are placed in between the single misaligned blades.

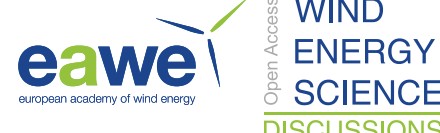

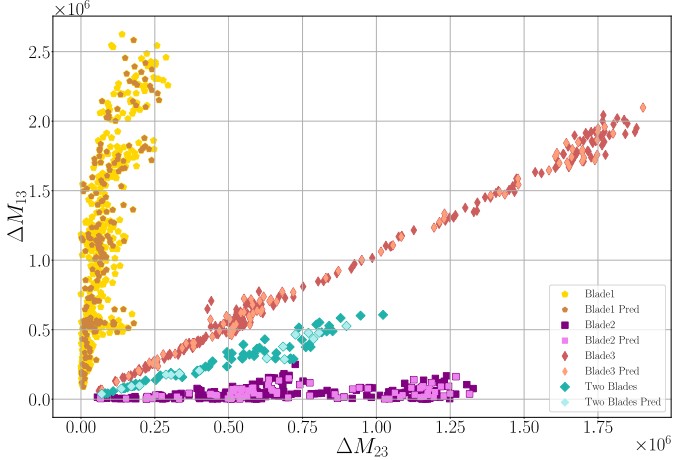

**Figure 10.** Fault isolation output, insights. In this figure, the case where *Blade* 2 and *Blade* 3 are simultaneously misaligned (light blue points in the plot) is reported. The lines are placed in between the single misaligned blades, hence *Blade* 2 (purple squares markers) and *Blade* 3 (red diamonds markers).

conditions by broadening the range of misalignment resolutions and wind degrees considered. This effort aims to further
evaluate the robustness of our approach and gain insights into its performance across a wider spectrum of conditions.

*Code and data availability.*  The machine learning-based tool performing pitch misalignment detection can be made available upon request. After review completion, the data used for generating all figures will be available for download through a data sharing platform, e.g. Zenodo.

*Author contributions.*  All authors provided fundamental inputs to this work through discussions, feedback, and analyses of the obtained results. S.M., J.L. and M.T. devised the main idea underlining the machine learning-based detector. S.C. and A.C. identified the physical
principles the detector algorithm is anchored on. S.C. and A.C. performed the multibody simulations of the wind turbine in healthy and unhealthy conditions. S.M., J.L. and M.T. implemented and performed the detection strategy.

*Competing interests.*  AC is member of the editorial board of *Wind Energy Science*.





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
