# Peer review of "A machine learning-based approach for active monitoring of blades pitch misalignment in wind turbines"

_Wind Energy Science, 2024_

## Referee Comment (RC3)

**Thank you for the opportunity to review this paper. Please find my suggestion for improving the manuscript below.**

**General Comments:**

- The abstract could be improved by stating more clearly the methodology followed and also giving a summary of the achieved main results and model accuracies.
- It should be made clear from the beginning that this is a simulation-based study, maybe by changing the wording from "experimental data" to simulation data.
- Since you introduce previous work and other studies on blade misalignment detection, your discussion section would benefit from putting your results in relation to prediction accuracies of other studies and modeling approaches.
- You mention that the methodology requires a minimal set of sensors, typically available in WT systems. As I understand your methodology relies on the x,y,z blade root bending moments. Can you comment on the availability of these sensors in real wind fleets and their reliability? How would issues like sensor drift affect your methodology?

**Specific Comments:**

Line 25: The differentiation between machine-learning and model-based methods is a bit confusing as machine-learning models can also be used to compare expected vs. actual turbine behavior. Consider differentiating between first principle models and machine learning models.

Line 37: Do you mean the models tend to be tailored to a specific turbine? Please clarify.

Line 55: What signals would that be? Naming them would make it easier to understand the difference in your approach.

Line 57: I suggest changing to third person perspective:

In (Milani et al., 2024), the potential of machine-learning methods for detecting pitch misalignment was recognized....

Line 63: Why is interpretability an issue if the method can "detect and locate misalignments, even in turbulent conditions, with satisfactory performance."?

Line 69: Naming the signals you use would improve clarity.

Line 145: This is the first time cp-lambda is mentioned. Introduce the tool in the beginning of the paragraph.

Paragraph 2.2: I think this paragraph lacks clarity. It should give a clear overview of the data simulation data set, including relevant input parameters and signals used for further analysis. The differentiation between "data sets" and "cases" is unclear. Also, the details on regression and classification analysis are not introduced yet and, therefore, difficult to understand.

Line 163: Spectral analysis?

Figure 2+3: Clearly state what can be seen in the upper and lower plot. Where is the 0.2 Hz peak? Is the unit on the y-axis correct?

Line 235: Have you chosen a smaller window than in the initial study? Please clarify.

Paragraph 3/ Figure 5: It is stated that both layers rely on random forest and regression models. Please clarify what each model in the different layers does. Also, the text could be better aligned with what is shown in the figure. What is "Strategy" referring to?

A more detailed explanation of the nature and functioning of the models would be beneficial. What kind of regression model is used, how is it tuned, etc?

Line 251: Restructure the sentence

Line 253: Do you use the same classification as in the previous study in the present study? Please clarify

3.1.1/3.1.2 Are these two separate approaches, or are they both used simultaneously? Please clarify also concerning Figure 5.

Line 284: Was a hyperparameter tuning process involved, or were hyperparameters selected based on the authors' experience?

Figure 7: Are the yellow points representing the reference value? Add to legend.

Figure 8: In line 296, you mention measuring the accuracy of the regression model in RMSEN, but here, absolute values are shown. Would it make sense to show RMSEN box plots?

Figure 9: Caption and text; please explain what can be seen in the subplots.

Figure 10: not mentioned in the text.

---

## Author Comment (AC1)

Dear Referee #1,

First and foremost, thank you for reviewing our paper *"A machine learning-based approach for active monitoring of blades pitch misalignment in wind turbines" (Preprint wes-2024-100)*. We are glad that your feedback was positive.

We have carefully revised the original manuscript to accommodate all your suggestions, taking this opportunity to make minor improvements to the text. In the amended version of the paper, the changes marked in orange refer to your comments. The point-by-point reply to your comments is reported here below.

Major comments

[Reviewer] Line 69: My one main comment/concern for this paper is the reproducibility. You mention that your method uses a "minimal set of sensors", but you never mention which sensors/measurements you use from the wind turbine. Without a clear discussion of what in-situ measurements are needed, this method will not be able to be reproduced or tested in the field.

[Answer] We agree with the reviewer. With the sentence "minimal set of sensors" we refer to the fact that the proposed methodology relies on a few sensors commonly installed on wind turbines for monitoring purposes. In particular, the measurements employed are 1) flap-wise and edge-wise blade root moments, sampled to capture the averaged values over a rotor revolution; 2) Nodding and yawing moments measured at main bearing, sampled to capture amplitude and phase of the oscillation at rotor frequency; 3) wind speed, sampled to capture averaged values over 10 minutes; 4) Rotor azimuth angle, sampled to perform the demodulation of nodding and yawing moments. No additional signal is required by the presented version of the approach to perform pitch misalignment detection and localization.

Additionally, we are actively working on improving the reproducibility of our approach, addressing the fact that moment sensors may not be available in some wind turbines. Accordingly, we recently submitted a paper focused on estimating the robustness of the presented approach when we use acceleration measurements, more often available, instead of loads, proving that similar performance is achieved in both the detection and localization of the fault. We will ensure to keep the reviewer informed should our submission be accepted.

[Action] As we agree with the Reviewer in considering this information key to appreciate the value of the work, we revised Section 2.2 "Collected dataset structure" in the manuscript accordingly, to better define the set of sensors and measurements needed to implement the proposed approach.

Minor comments

[Reviewer] Line 15-20: You may want to add a brief discussion about what typical misalignment ranges are and how frequently this occurs on a given turbine or within an array.

[Answer] Yes. Such a discussion is already present in literature (e.g. Astolfi, Machines 2019,

7(1), 8; https://doi.org/10.3390/machines7010008, and Kusiak and Verma 2011 IEEE Trans. Sustain. Energy1 87–96, being this latter already mentioned in our literature review). Hence, we will add some sentences, as suggested by the Reviewer, referring to the existing literature.

[Action] The text has been modified according to what we previously stated, specifically in the Introduction (Section I).

*[Reviewer] Does turbine misalignment decrease energy production? .*

[Answer] Yes, especially in partial power region, but it strongly depends on the severity of the problem. Mild misalignment, which may create significant vibratory problems, may have a limited impact on power and annual energy production. We can easily quantify such a decrease and we can provide a picture (or a table) of that.

[Action] We included a discussion on pitch misalignment-induced power in Section 2.2 of the revised version of the manuscript, showing the impact of pitch misalignment on power production.

*[Reviewer]Also, is there a significant threshold where turbine misalignment becomes more of an issue/concern?.*

[Answer] According to Regulation, the design of wind turbines is subject to the presence of a small misalignment equal to 0.3 deg. Hence, a good detection strategy should be able to capture misalignment of at least 0.3 deg to maintain the machine within its operative limits.

[Action] This discussion is added to the Introduction and to the Conclusions of the manuscript.

*[Reviewer] Line 147: Would it be beneficial to show the power curve here?*

[Answer] We agree with the Reviewer but, instead of showing a picture, it is better to simply report the values of cut-in, cut-out and rated speeds.

[Action] The text has been modified accordingly.

*[Reviewer] Line 155: What ranges of wind speeds are you considering? Is there a difference in algorithm performance when the turbine is just above cut-in vs. at rated capacity?*

[Answer] The velocities considered in this work range from cut-in velocity, $v = 3m/s$, to cut-out velocity, $v = 25m/s$. For example, when considering the algorithm performance only at cut-in velocity, the F1-score for the first classification layer is 65%. On the other hand, at $13m/s$, the

F1-score of the first layer is 90.7%. Indeed, at lower wind speeds performance is reduced since the higher turbulence intensity leads to a more difficult classification. However, the results remain well above 50% confirming the robustness of the approach even in these challenging conditions.

[Action] We have explicitly stated the values for cut-in, rated and cut-out speeds in the paper, without including any additional comments on the performance at selected speeds.

*[Reviewer] Figures 2 and 3: I would recommend adding (a) and (b) to each figure. It will improve clarity when references the figures in the text.*

[Answer] We agree with the Reviewer.

[Action] Figure 3 in the revised version of the manuscript has been modified according to the Reviewer's suggestions, Figure 4 is a stand-alone figure.

We look forward to your kind reply, and in the meanwhile, we send our warmest regards.

Sincerely yours,

Sabrina Milani, on behalf of all Authors.

---

## Author Comment (AC2)

Dear Referee #2,

First and foremost, thank you for reviewing our paper *"A machine learning-based approach for active monitoring of blades pitch misalignment in wind turbines" (Preprint wes-2024-100)*. Thank you for your positive feedback.

We have revised the original manuscript to include all your suggestions. In the amended version of the paper, the changes marked in red refer to your suggestion. The reply to your major comment is reported here below.

Major comment

*[Reviewer] I noticed that the manuscript focuses on the scenarios with unbalanced blade misalignments. These unbalanced cases ensure the Yawing moments differ from the healthy case. I was wondering if the Yawing moments will still be at the non-N Rev positions when the three turbine blades have the same amount of misalignments. In this case, can the proposed ML model make accurate predictions?*

[Answer]
The cases in which all blades present the same amount of misalignment, which can be briefly called "collective misalignment", were analyzed but not included in the manuscript. The collective misalignment does not imply an imbalanced rotor, because the rotor loads remain almost symmetrical, with the wind turbulence being the only mild source of asymmetric loading. Consequently, the characteristic peak at 1xRev is not visible in the spectral analysis. Regarding the ML performance in these cases, since the vibrations do not present distinguishing features compared to the healthy case, the model would not detect significant anomalies in terms of yawing and nodding moments.

To also detect collective misalignment, the ML model could be integrated with additional features, e.g. power data. Preliminary investigations show that, in the above-rated speed range, the controller compensates for the collective misalignment and trims the machine at maximum power, basically canceling out its fingerprint in the produced power. However, at low wind speeds (below rated), the collective misalignment is associated with a mild reduction of power with respect to the healthy case. Since we haven't yet obtained consolidated results, we prefer not to update the present methodology, leaving this opportunity as a future application.

[Action] No additional analysis on this specific scenario has been included in the manuscript, as the focus was on unbalanced cases where rotor asymmetries lead to more prominent yawing and nodding moments and detectable anomalies. A sentence, to briefly describe the case of collective misalignment, is included in the conclusion within the paragraph related to possible outlooks.

We look forward to your kind reply, and in the meanwhile, we send our warmest regards.

Sincerely yours,

Sabrina Milani, on behalf of all Authors.

---

## Author Comment (AC3)

Dear Referee #3,

First and foremost, thank you for reviewing our paper *"A machine learning-based approach for active monitoring of blades pitch misalignment in wind turbines" (Preprint wes-2024-100)*. Thank you for all of your comments.

We have carefully revised the original manuscript to accommodate all your suggestions, taking this opportunity to make minor improvements to the text. In the new version of the paper, the changes related to your suggestions are marked in purple. The point-by-point reply to your comments is reported here below.

Major comments

*[Reviewer] The abstract could be improved by stating more clearly the methodology followed and also giving a summary of the achieved main results and model accuracies.*

[Answer] We agree with the Reviewer

[Action] We revised the abstract in the manuscript according to the Reviewer's suggestion, to better define the methodology and include main results and model accuracy.

*[Reviewer] It should be made clear from the beginning that this is a simulation-based study, maybe by changing the wording from "experimental data" to "simulation data".*

[Answer] The Reviewer was right. In fact, as stated from the beginning of our manuscript (as in Line 13 "The approach is validated across an extended operational envelope using data gathered **from a state-of-the-art simulation model** commonly used for designing and certifying commercial wind turbine systems") the exploited data in the classification architecture comes from a simulation environment and no experimental data are exploited in the manuscript.

[Action] We revised all misleading lines by changing "experimental" to "simulation" data.

*[Reviewer] Since you introduce previous work and other studies on blade misalignment detection, your discussion section would benefit from putting your results in relation to prediction accuracies of other studies and modeling approaches*

[Answer] The previously cited work (Milani et al., 2024), on which the current manuscript relies, focused on a simplified scenario. Specifically, classification was performed on single misaligned blades, without considering localization issues and using a reduced range of blade misalignment (starting from 0.5 degrees) and excluding the regression approach. While we could mention the prediction accuracy and performance of the algorithm in that context, it is important to note that the simulation data and the scenario considered in the current manuscript are more complex. In this work, we have extended the analysis to include multiple blades with misalignments and larger ranges of misalignment angles (starting from 0.1 degrees). Additionally, we have addressed the localization procedure, which was not considered in the previous study. As the same applies to all other works in the literature that focus solely on pitch misalignment detection under simplified conditions, a fair comparison cannot be provided.

[Action] We do not foresee any action as we believe that the current state of the text is satisfactory.

*[Reviewer] You mention that the methodology requires a minimal set of sensors, typically available in WT systems. As I understand your methodology relies on the x,y,z blade root bending moments. Can you comment on the availability of these sensors in real wind fleets and their reliability? How would issues like sensor drift affect your methodology?*

[Answer] As the Reviewer correctly pointed out, our methodology relies on blade root bending, nodding and yawing Moments, which are essential inputs for detecting misalignment. These signals are measured by sensors commonly available in many modern wind turbines, especially in larger ones for structural monitoring purposes. Regarding their reliability, these sensors are generally robust and designed for long-term operation under harsh environmental conditions. However, sensor drift and degradation over time could pose challenges. Sensor drift, if left uncorrected, could introduce errors in the measurements and potentially affect the accuracy of the machine-learning model. To mitigate such issues, the methodology could be additionally integrated with sensor health monitoring and calibration techniques to account for drift. For instance, periodic re-calibration of the sensors could help detect and correct sensor anomalies, ensuring that the model continues to make accurate predictions despite sensor drift. Additionally, redundancy in sensor systems, where multiple sensors provide overlapping data, could enhance the robustness of the methodology against individual sensor failures or degradation. This extension is certainly interesting but falls out of the scope of the present paper.

[Action] We are actively working on improving the reproducibility of our approach, addressing the fact that moment sensors may be not available in some wind turbines. Accordingly, we recently submitted a paper focused on estimating the robustness of the presented approach when we use acceleration measurements, more often available, instead of loads, proving that similar performance is achieved in both the detection and localization of the fault. We will ensure to keep the reviewer informed should our submission be accepted. The conclusion section is also updated to point out this.

Minor comments

*[Reviewer] Line 25: The differentiation between machine-learning and model-based methods is a bit confusing as machine-learning models can also be used to compare expected vs. actual turbine behavior. Consider differentiating between first principle models and machine learning models.*

[Answer] We agree with the Reviewer that this differentiation may be not clear in the original version of the manuscript. We revise the text by differentiating more explicitly between first-principle models (i.e., model-based), which rely on the physical equations governing turbine behavior, and machine learning models (i.e., model-free), which learn from data to make predictions without relying on explicit physical laws.

[Action] Model-based approach has been substituted with first-principle models in the Introduction.

*[Reviewer] Line 37: Do you mean the models tend to be tailored to a specific turbine? Please clarify.*

[Answer] Yes, in a model-based approach, a physical model of the specific turbine under consideration would be required. This model is typically fine-tuned for each individual turbine and may need to be adjusted for turbines with different characteristics. Moreover, approaches based on first-principle models may be also prone to error due to the simplifications made in the development of the mathematical models.

[Action] We do not believe that the text should be changed.

*[Reviewer]Line 55: What signals would that be? Naming them would make it easier to understand the difference in your approach.*

[Answer] The signals referenced in the cited works primarily refer to force signals, such as the aerodynamic forces acting on the blades or structural forces in the turbine components which are rarely available on a real system. In contrast, our approach includes more available signals, such as blade root bending moments, which provide a more comprehensive view of the turbine structural behavior and allow for more accurate detection of anomalies related to blade misalignment in real systems.

[Action] The text has been modified according to the Reviewer's suggestion to include examples of signals used in literature-based approaches that are often difficult to obtain in real-world turbines.

*[Reviewer] Line 57: I suggest changing to third person perspective: In (Milani et al., 2024), the potential of machine-learning methods for detecting pitch misalignment was recognized....*

[Answer] We agree with the Reviewer.

[Action] The text has been modified as sugested.

*[Reviewer]Line 63: Why is interpretability an issue if the method can "detect and locate misalignments, even in turbulent conditions, with satisfactory performance."?*

[Answer] While it is true that the method can effectively detect and locate misalignments, even in turbulent conditions, accuracy cannot be the only metric to rely on for evaluating the performance of an anomaly detection approach. Indeed, interpretability remains a critical issue for several reasons. Complex machine learning models, like neural networks, can often be treated as

"black boxes", making it difficult to understand why the model makes certain predictions. This is important because, in critical systems like wind turbines, operators should not only rely on predictions but need to also understand the reasoning behind them, especially when anomalies are detected.

Additionally, interpretability is crucial when it comes to diagnosing and troubleshooting. If the model can pinpoint an anomaly but the reasoning behind it is unclear, it could be challenging to determine the root cause and take corrective actions. For example, knowing that a misalignment is detected is useful, but understanding how the different input signals (e.g., blade root bending moments) contributed to this detection can guide more informed maintenance decisions. For this reason, to address this, our approach prioritizes interpretability by incorporating physics-based features and utilizing explainable machine learning models like linear regression and random forest.

[Action] We added a few lines to the text at the end of the paragraph "Machine-learning approaches" to stress this fact.

*[Reviewer] Line 69: Naming the signals you use would improve clarity.*

[Answer] We agree with the Reviewer.

[Action] We have explicitly mentioned the signals exploited in our study.

*[Reviewer] Line 145: This is the first time cp-lambda is mentioned. Introduce the tool in the beginning of the paragraph.*

[Answer] The reviewer is right. The tool is already introduced at the beginning of the paragraph, ("To achieve this, we ... a high-fidelity simulation multibody tool (Bottasso and Croce, 2009–2018).", but its name is not declared.

[Action] We added the name of the tool at the beginning of section 2.2, as suggested by the reviewer.

*[Reviewer] Paragraph 2.2: I think this paragraph lacks clarity. It should give a clear overview of the data simulation data set, including relevant input parameters and signals used for further analysis. The differentiation between "data sets" and "cases" is unclear. Also, the details on regression and classification analysis are not introduced yet and, therefore, difficult to understand.*

[Answer] We agree with the Reviewer.

[Action] We have revised the text to clarify the distinction between 'dataset' and 'cases,' providing more detailed explanations of our intended meanings for these terms. Moreover, the dataset used for the development of this work, i.e. all outputs of `Cp-Lambda` simulations, will be made

available through an open data-sharing platform, such as Zenodo.

*[Reviewer] Line 163: Spectral analysis?*

[Answer] We agree with the Reviewer

[Action] We have changed the text accordingly.

*[Reviewer] Figure 2+3: Clearly state what can be seen in the upper and lower plot. Where is the 0.2 Hz peak? Is the unit on the y-axis correct?*

[Answer] The reviewer is right. As indicated in the legend of the figure, we gave proof that the expected peak induced by the misalignment happens at different frequencies according to the rotor speed. However, if we perform the spectral analysis with respect to the azimuth signal, the peak remains centered around the $1 \times$ Rev.

[Action] We modified the figures to improve clarity.

*[Reviewer] Line 235: Have you chosen a smaller window than in the initial study? Please clarify.*

[Answer] As stated in the text in Line 234-235, the window of the sliding factor has been set to one rotor revolution. The last comment "A smaller sliding factor, such as one revolution..." is a statement added to provide the reader with an intuition of the effect of varying that parameter. However, we would like to note that the sliding factor does not impact classification performance. Its sole use is to determine the frequency of prediction update.

[Action] No action is considered.

*[Reviewer] Paragraph 3/ Figure 5: It is stated that both layers rely on random forest and regression models. Please clarify what each model in the different layers does. Also, the text could be better aligned with what is shown in the figure. What is "Strategy" referring to? A more detailed explanation of the nature and functioning of the models would be beneficial. What kind of regression model is used, how is it tuned, etc?*

[Answer] At the beginning of the section in Line 238 the following statements are reported: "This Section outlines the hierarchical classification architecture we designed for detecting and isolating pitch misalignment. As described in Section 1, our classification pipeline features two layers: the first identifies misalignment, categorizing its severity as low, medium, or high using the classification model, or precisely predicting the amount using the regression one, based on

user preferences. The second layer, given that a misalignment is detected, localizes the affected blades. Both layers rely on random forest models for classification, and linear regression for the regression. These models have been chosen for their robustness, accuracy, and interoperability. Specifically, this algorithm combines the output of multiple shallow binary tree classifiers, which splits data according to specific criteria, minimizing the differences in the child nodes at each iteration."

Indeed, we are considering an approach in which the first layer aims to identify and classify the misalignment, relying on the classification or on the regression approach depending on the available training data size and to the end application requirements; then, the second layer aims to localize the misaligned blade.

In more detail, for the classification, one between two different strategies can be followed:

- the first one relies on a Random-Forest classifier that categorizes the misalignment entity in four classes healthy or low, medium, or high misalignment.
- the second one relies on a Linear Regression classification model, that is capable of precisely predicting the specific amount of misalignment instead of producing a 4 classes classification output.

For the localization, random forest has been employed.

As for the description of the employed models and the tuning of their parameters, section 3.1.2 provides details about the two models.

[Action] The arrow between the two strategies has been removed in Figure 5 to underline that the two strategies are comparable, so one between them can be adopted.

*[Reviewer] Line 251: Restructure the sentence*

[Answer] We agree with the Reviewer.

[Action] We have restructured the sentence.

*[Reviewer] Line 253: Do you use the same classification as in the previous study in the present study? Please clarify*

[Answer] In the previous study, only the Anomaly Detection layer has been applied to the data. Specifically, we used the Random-Forest classifier strategy on a reduced set of data where misalignment was related to a single blade and with misalignment starting from $0.5 degree$. In the current paper, we extended our dataset to include scenarios with multiple misaligned blades, demonstrating that the approach still achieves high performance. Additionally, we enhance the original framework architecture by designing a hierarchical classification structure. After the initial layer, aimed at reporting the presence of a misalignment, our novel methodology integrates an additional layer comprising a random forest classifier to localize the specifically affected blade in case misalignment is detected.

[Action] No action is considered.

*[Reviewer] 3.1.1/3.1.2 Are these two separate approaches, or are they both used simultaneously? Please clarify also concerning Figure 5.*

[Answer] Yes, indeed these are two separated approaches. The Misalignment Classification relates to the previously mentioned first strategy, where the classification is a 4 class outputs (the one employed in the previous work as well), while the Misalignment Detection, as mentioned in the text, is the expansion of the previous work, that consists of the introduction of a second and comparable strategy, namely the Linear Regression, that is also explained in Figure 5. As a matter of fact, in line 262 the following sentences are reported: "In this study, we expand upon previous research by introducing and evaluating the performance of a linear regression model. This model, trained using the same set of features employed in the misalignment classification, offers precise quantification of misalignment severity considering default hyper-parameters."

[Action] Again, the arrow between the two strategies has been removed in Figure 5 to underline that the two strategies are comparable.

*[Reviewer] Line 284: Was a hyperparameter tuning process involved, or were hyperparameters selected based on the authors' experience?*

[Answer] As mentioned in the text, hyperparameters were set to their default values unless otherwise specified. Specifically, for the Random Forest classifiers, the depth and number of estimators hyperparameters were fine-tuned based on the data, while for the linear regression model, default parameters were used.

[Action] No action is considered.

*[Reviewer] Figure 7: Are the yellow points representing the reference value? Add to legend.*

[Answer] The yellow points represent as in the previous figure, the predicted instances.

[Action] The updated legend has been added to the plot.

*[Reviewer] Figure 8: In line 296, you mention measuring the accuracy of the regression model in RMSEN, but here, absolute values are shown. Would it make sense to show RMSEN box plots?*

[Answer] Sorry for the misunderstanding, in line 296 we mention measuring the accuracy of the regression model in RMSE, not RMSEN.

[Action] No action considered.

*[Reviewer] Figure 9: Caption and text; please explain what can be seen in the subplots.*

[Answer] The description of the plots is reported in Line 320-327. Additional comments on the plots will be added in the caption.

[Action]Additional comments have been added in the caption.

*[Reviewer] Figure 10: not mentioned in the text.*

[Answer] We agree with the Reviewer.

[Action] Figure 10 has been mentioned in the text, when opportune.

We look forward to your kind reply, and in the meanwhile, we send our warmest regards.

Sincerely yours,

Sabrina Milani, on behalf of all Authors.